# Prognostic Value of Peritoneal Cytology in Stage I Serous and Clear Cell Carcinoma of the Endometrium

**DOI:** 10.3390/jcm12041609

**Published:** 2023-02-17

**Authors:** Jie Yang, Jiaxin Yang, Dongyan Cao, Ming Wu, Yang Xiang

**Affiliations:** Department of Obstetrics and Gynecology, Peking Union Medical College Hospital, National Clinical Research Center for Obstetric & Gynecologic Diseases, Beijing 100730, China

**Keywords:** uterine serous carcinoma, uterine clear cell carcinoma, peritoneal cytology, recurrence, survival outcomes

## Abstract

(1) Background: To investigate the relation between malignant peritoneal cytology and survival outcomes in patients who underwent primary staging surgery for stage I uterine serous (USC) or clear cell carcinoma (UCCC). (2) Methods: In this retrospective analysis, patients with stage I USC or UCCC who underwent staging surgery between 2010 and 2020 at the Peking Union Medical College Hospital were identified and reviewed. (3) Results: A total of 101 patients were included, and 11 patients had malignant cytology (10.9%). The median follow-up time was 44 months (range 6–120) with a total of 11 (10.9%) recurrences. Patients with malignant cytology had a higher likelihood of peritoneal recurrence and a shorter time to relapse (13 vs. 38 months, *p* = 0.022), as compared to patients with negative cytology. In univariate analysis, malignant cytology and serous histology had worse progression-free survival (PFS) and overall survival (OS) (all, *p* < 0.05). In sensitive analysis, the detrimental effects of malignant cytology on survival were more prominent in patients over 60 years old, those with serous histology, stage IB disease, and those who received hysteroscopy as a diagnostic test. (4) Conclusions: Stage I USC or UCCC patients with malignant peritoneal cytology had a higher recurrence and inferior survival.

## 1. Introduction

Endometrial cancer (EC) is the most common gynecologic malignancy in developed countries [1]. Endometrial malignancies have been divided into two subtypes based on clinicopathologic features [2]. Type I tumors have endometrioid histology, comprising approximately 80% of EC, and are generally diagnosed in the early stages with a favorable prognosis. On the contrary, type II ECs, including serous, clear-cell, undifferentiated, high-grade endometrioid carcinoma, etc., are associated with a poorer prognosis compared to type I ECs [2]. Since the malignant behavior of these two types of EC is varied, the management of each is different in clinical practice. In type II ECs, the prognostic factors in patients with stage I disease are even more important because post-operative adjuvant therapy in patients with certain risk factors may significantly reduce the rate of tumor relapse, leading to better survival outcomes [3,4].

The prognostic value of malignant peritoneal cytology is controversial. Previous studies have not been able to identify a significant association between cytology and prognosis in early-stage and low-risk EC [5,6]. The 2009 revision of the cancer staging system by the International Federation of Gynecology and Obstetrics (FIGO) has excluded malignant peritoneal cytology as a criterion for staging in ECs, and is not considered a factor for post-operative treatments [7]. The incidence of malignant peritoneal cytology is reported to be higher in type II EC cancer, but studies specifically evaluating malignant peritoneal cytology in stage I non-endometrioid endometrial cancer are limited [8,9]. Since the pathophysiologic features of type II EC are different from endometrioid types, evaluating the prognostic value of factors such as peritoneal cytology would be substantial.

The primary objective of this study was to evaluate the prognostic value of malignant peritoneal cytology in patients with stage I uterine serous cancer (USC) and clear cell cancer (UCCC).

## 2. Materials and Methods

The Institutional Review Board of Peking Union Medical College Hospital approved this retrospective study. We reviewed the clinical records of patients with stage I serous or clear-cell EC who underwent peritoneal cytology examination at staging surgery between 2010 and 2020.

The following information was extracted from the medical records: age at time of cancer diagnosis, year of diagnosis, initial diagnostic method in biopsy (dilation and curettage, D&C, versus hysteroscopy), histology subtypes (serous, clear cell), cancer stage (IA versus IB), tumor size (≤2 cm, 2.1–4 cm, >4 cm), lymph vascular space invasion (LVSI), depth of invasion (submucous, ≤1/2 myometrium, >1/2 myometrium), and peritoneal cytology results (malignant versus negative). The grade was not specially reviewed because USC and UCCC were regarded as high-grade endometrial cancer.

Surgical data reviewed included the procedure type (minimally invasive versus open surgery), performance of pelvic lymphadenectomy (LND) (yes versus no), appendectomy (yes versus no), and omentectomy (yes versus no). Post-operative treatments were documented as either chemotherapy, radiotherapy (vaginal brachytherapy, external beam radiotherapy [EBRT]), or both. The cytology status from surgical staging was reviewed and confirmed by two pathologists independently. The presence of malignant cells in the peritoneal cytology was defined as positive.

After treatment completion, patients were followed every 3–6 months for the first 2 years, 6–12 months for 3–5 years, then annually thereafter. Recurrence was confirmed by either imaging or surgical resection. The pattern of recurrence was documented as peritoneal (vaginal cuff, pelvis, or abdomen), lymph node metastasis (retroperitoneal, or above para-aortic nodes), hematogenous (liver, lung, bone, or brain), and multiple. Progression-free survival (PFS) was defined as the time (months) from the initial diagnosis to disease recurrence. Overall survival (OS) was defined as the time (months) from diagnosis to death from all causes. Data on patients with no evidence of disease recurrence or death were censored at the date of the last follow-up.

The Mann–Whitney U-test was used to analyze continuous variables as appropriate. Frequency distributions were compared using chi-square and Fisher’s exact tests for categorical variables. Baseline clinicopathological characteristics were compared between the malignant and negative peritoneal cytology groups. Survival outcomes (PFS and OS) were estimated using the Kaplan–Meier method and log-rank test. Univariate and multivariate Cox proportional hazard ratio (HR) analyses were performed to identify independent prognostic factors. The clinicopathological factors included in the univariate analyses were primarily selected from the commonly known prognostic factors by reviewing previous studies that investigated the prognostic factors in EC, including age, initial surgery type, stage, mitotic counts, and adjuvant chemotherapy. Possible prognostic factors were assessed using univariate analyses. Multivariate Cox models included factors associated with a statistically significant increased risk for recurrence or death in the univariate analysis. Stratified sensitivity analyses were performed to assess the association of peritoneal cytology results and survival outcomes by age (≤60 vs. >60 years), histology (serous, clear cell), cancer stage (IA vs. IB), diagnostic methods (D&C vs. hysteroscopy), and post-operative chemotherapy (yes vs. no). In all cases, a *p*-value < 0.05 was considered statistically significant. SPSS ver. 23.0 (IBM, Armonk, NY, USA) and Prism 6.0c software (GraphPad, San Diego, CA, USA) were used for statistical analyses.

## 3. Results

From January 2010 to December 2020, 101 patients with stage I serous or clear cell EC who underwent EC staging surgery in the gynecologic department of Peking Union Medical College Hospital were identified and included in the study.

The mean age of the patients was 61.6 (SD 10, range 31–81) years. The average pre-surgical tumor maximum diameter on imaging examination was 2.7 (SD 1.76, range 0.4–9) cm. A total of 11 patients had malignant peritoneal cytology (10.9%). There were 55 patients with serous carcinoma histology and 46 patients with clear cell carcinoma included in this study. More than half of the patients had earlier-stage disease (IA 77 patients, IB 24 patients). Lymph-vascular space invasion (LVSI) was found in 10 patients (9.9%). The initial diagnostic procedure was either dilation and curettage (D&C) (*n* = 51, 50.5%) or hysteroscopy (*n* = 50, 49.5%). There were 63 patients (62.4%) who underwent open surgery and 38 patients who underwent laparoscopic surgery (37.6%). Omentectomy was performed in 86 (85.1%) patients and appendectomy in 36 (35.6%) patients. Lymphadenectomy was omitted in 7 patients due to advanced age or other severe comorbidities. Table 1 summarizes patient demographics and tumor characteristics allocated by cytology result. None of these factors (age, histology type, tumor size, depth of invasion, LVSI, initial diagnostic surgery type, route of staging surgery, performance of omentectomy, and lymphadenectomy) were associated with malignant peritoneal cytology.

In the 11 patients with malignant peritoneal cytology, 8 patients were diagnosed with serous carcinoma and 3 patients were diagnosed with clear cell carcinoma. Among the 8 patients with serous carcinoma, 4 patients underwent open surgery, and 4 patients underwent MIS; 5 of them received adjuvant chemotherapy of 3 cycles of TC. Among the 3 patients with clear cell carcinoma, 2 patients received open surgery and 1 received MIS; they all received adjuvant chemotherapy of 3 cycles of TC.

The post-operative management included chemotherapy and radiotherapy. There were 53 patients (52.4%) who received chemotherapy alone. Radiation therapy after adjuvant chemotherapy was performed in 12 patients (11.9%). Only one patient received EBRT and brachytherapy without chemotherapy due to impaired kidney function. The radiation therapy included EBRT in 2 patients (1.98%), 8 vaginal brachytherapy (7.92%), and 3 in both EBRT and brachytherapy (2.97%). There were no differences in preference for adjuvant treatments of chemotherapy (*p* = 0.74) or radiation therapy (*p* = 0.49) between the cytology-positive and negative groups. The prognostic factors in patients who underwent adjuvant chemotherapy were similarly distributed between the cytology-positive group and negative group: in the cytology-positive group, there were 6 patients diagnosed with serous cancer and 2 patients with clear cell cancer, compared to 34 and 23 patients in cytology negative group, respectively (*p* = 0.65); there were 5 patients had tumor size less than 2 cm and 3 patients had tumor size over 2 cm in cytology positive group, as compared to 35 and 22 in cytology negative group, respectively (*p* = 0.77); there were 3 patients had deep myometrium invasion in cytology positive group, as compared to 13 patients in cytology negative group, respectively (*p* = 0.64). Thus, we consider the basic characteristics between the two groups to be balanced.

The median follow-up time was 44 months (range 6–120). A total of 11 (10.9%) patients developed recurrence: 6 (54.4%) in the malignant cytology group and 5 (5.6%) in the negative cytology group. In the malignant cytology group, there was 1 patient with recurrence at the vaginal cuff, 2 patients with pelvic recurrence, 2 patients with multiple tumor implantation in the abdomen, and 1 patient with lung metastasis. In the negative cytology group, there was 1 patient with retrorectal space recurrence, 2 patients with pelvic and para-aortic lymph node recurrence, and 2 patients with lung metastases. Patients with malignant cytology had a higher peritoneal recurrence (5/6, 83.3%) vs. patients with negative cytology (1/5, 20%) (*p* = 0.036). The time to relapse was shorter in the malignant cytology group (median 13, range 3–41 months) as compared to patients with negative cytology (median 38, range 15–95 months) (*p* = 0.022) (Table 2).

The demographic features between the cytology positive and negative groups were not significantly different, thus the survival outcomes were compared between women with malignant and negative peritoneal cytology. Patients with malignant cytology had inferior PFS and OS as compared to those who had negative cytology. The 5-year PFS and OS were 39.8% and 60.6% for the malignant cytology group, and 94.2% and 97.8% for the negative cytology group, respectively. Malignant peritoneal cytology was associated with nearly a 12-fold increased risk of recurrence (HR 12.6, 95% CI 1.54–103, *p* < 0.001) and a 9-fold increased risk of all-cause mortality (HR 9.66, 95% CI 1.21–112, *p* < 0.001) (Figure 1).

Univariate analysis of other potential prognostic factors for survival outcomes was performed. On PFS analysis, histology type of USC (HR 4.56, *p* = 0.031) was a poor prognostic factor. Advanced age (HR 6.95, *p* = 0.030), higher stage (HR 12.5, *p* = 0.009), and no omentectomy at surgery (HR 7.96, *p* = 0.015) were associated with decreased OS. Significant variables from univariate analysis were included in multivariate analysis. On multivariate analysis, only malignant cytology had statistical significance in predicting unfavorable PFS. When analyzing OS, malignant cytology, higher stage, and no omentectomy were found to be poor prognosticators (Table 3).

In sensitivity analyses, malignant cytology was associated with decreased PFS in patients over 60 years old (HR 19.05), who had serous histology (HR 24.12), stage IB (HR 2063), and chemotherapy after surgery (HR 10.78). Malignant cytology was associated with inferior OS in patients who had serous histology (HR 20.26) and post-operative chemotherapy (HR 23.77). The difference in PFS and OS between the patients with or without malignant peritoneal cytology was most significant for patients in the stage IB subgroup (Figure 2). Diagnostic hysteroscopy seems more hazardous in patients with malignant cytology than D&C in Figure 2, thus a subgroup analysis on hysteroscopy was performed. In patients with malignant cytology, hysteroscopy was associated with inferior PFS (HR 5.48, 95% CI 1.47–33.4, *p* = 0.03) and OS (HR 6.32, 95% CI 0.96–51.5, *p* = 0.05), as compared to D&C. In the negative cytology cases, there were no survival differences in PFS (*p* = 0.88) or OS (*p* = 0.20) between patients who received diagnostic hysteroscopy or D&C.

## 4. Discussion

In this study, we found that malignant peritoneal cytology was independently associated with inferior PFS and OS in patients with stage I USC and UCCC. Malignant peritoneal cytology was seen in 10.9% of stage I USC and UCCC patients. Patients with malignant peritoneal cytology had more peritoneal recurrence and a shorter time to relapse.

The incidence of malignant peritoneal cytology in stage I EC has been reported from 5.3% to 10.5% in prior studies, and it is higher in non-endometrioid EC [10,11,12]. A retrospective observational study using the Surveillance, Epidemiology, and End Results (SEER) database reported a 9.2% incidence of malignant peritoneal cytology in non-endometrioid EC and 10.7% in USC/UCCC [11]. We found a similar incidence of 10.9% in USC/UCCC.

The most recognized theory for the mechanism of malignant peritoneal cytology is the retrograde spread of tumor cells via the fallopian tubes. A review of 4489 EC patients and their history of tubal ligation revealed that tubal ligation is inversely associated with stage III/IV disease [13]. The risk reduction of peritoneal metastasis is much more significant in patients with serous histology than in endometroid cancer (OR 0.28, 95% CI 0.11–0.68 vs. OR 1.27, 95% CI 0.41–3.89) [13]. Non-endometrioid EC is more likely to have trans-tubal spread, which may explain the higher incidence of malignant peritoneal cytology in USC/UCCC than endometrioid tumors.

We compared the clinical features between the malignant and negative peritoneal cytology groups, and none of the clinical and pathological factors were identified as independent risk factors for malignant cytology in stage I USC/UCCC. One study reported that diagnosis at an older age (>78 years old), serous histology, and larger tumor size were associated with increased risks of malignant peritoneal cytology [11]. Other studies, however, have failed to find an association between clinical-pathological factors and malignant peritoneal cytology [12,14]. A meta-analysis demonstrated that hysteroscopy is associated with a higher rate of positive peritoneal cytology when compared with no hysteroscopy, but this phenomenon is not identified in early-stage EC [15]. For patients with postmenopausal bleeding, hysteroscopy was considered more sensitive and accurate as compared to endometrial sampling and blind dilation and curettage [16]. It was reported that blind dilation and curettage can miss nearly 50% of the uterine cavity’s endometrium, in terms of a high false-negative rate. Since the prognostic value of positive cytology was minimal, especially in early-stage EC, and a lack of evidence that hysteroscopy could cause tumor cell influx into the abdominal cavity, hysteroscopy was used as the gold standard in the diagnosis of abnormal uterine bleeding in our institution. In our study, we did not find a relationship between hysteroscopy and positive peritoneal cytology. However, in patients in the malignant peritoneal cytology subgroup, we found hysteroscopy to be associated with inferior survival outcomes. Patients with malignant peritoneal cytology may have tumor cells more capable of migration and invasion. The intrauterine pressure during hysteroscopy can lead to tumor cell reflux and intra-abdominal dissemination. In our institution, we are becoming more conservative about using hysteroscopy as a diagnostic method if we suspect EC.

In our study, we found a higher recurrence rate and shorter time to relapse in patients with malignant peritoneal cytology as compared to those who have negative cytology. The recurrence patterns were different between the two groups. A retrospective study in low-risk EC including 478 patients reported an increased intra-abdominal recurrence in patients with positive peritoneal cytology [17]. Our study also suggested that patients with malignant cytology were more likely to have a peritoneal recurrence.

The strengths of this study include: a collection of patients from one, large academic medical institution. This reflects the standard surgical approaches and postoperative management of our practice; data were complete and well documented in all patients with consistent pathology review; the long follow-up period; and the balanced basic characteristics between groups.

Our study has certain limitations: first, inherent bias given the retrospective nature of the study; second, the limited number of patients with malignant peritoneal cytology resulted in a few underpowered analyses; and third, highly individualized postoperative treatment added heterogeneity to this study.

The prognostic importance of malignant peritoneal cytology in early EC is controversial [9,12,18]. In the revised 2009 FIGO staging system, positive peritoneal cytology is no longer included in defining surgical staging [19]. Several large retrospective studies using the SEER database [10] and the National Cancer Database (NCDB) [9] have reported that positive peritoneal cytology is significantly associated with decreased survival in early-stage EC. Historical arguments for the omission of peritoneal cytology in the staging of EC were predominantly focused on type I EC. Type II EC has a worse overall survival and higher recurrence rate, and the rate of malignant peritoneal cytology is much higher than type I EC [11]. The recognition of peritoneal cytology as a tumor prognostic factor is necessary for early-stage non-endometrioid EC.

The role of cytology as a prognostic indicator in type II EC is under-evaluated [8]. Our study demonstrated that malignant peritoneal cytology was associated with decreased PFS and OS in early-stage USC/UCCC, and the impact on survival is quite significant (37% decrease in 5-year OS). Multivariate analysis in our study also suggested that malignant peritoneal cytology was an independent risk factor for both PFS and OS. These findings are consistent with two previous reports focusing on non-endometrioid EC [11,14]. Matsuo et al. reported that the mortality risk is nearly tripled (HR 2.59) with a 5-year survival rate decrease of 19.4% in stage I non-endometrioid EC patients with malignant peritoneal cytology [11]. Another case-series study focusing on USC showed that in patients with intermediate- and high-risk early-stage USC, cytology status is significantly related to prognosis [14].

The surgical staging for type II EC in the National Comprehensive Cancer Network (NCCN) guidelines includes TH/BSO, peritoneal lavage for cytology, and omental and peritoneal biopsies [20]. We found omentectomy was related to superior OS, but not PFS. This unpaired outcome may be due to the frail physical status of the patients who did not receive omentectomy, such as advanced age or medical comorbidities. Physicians may omit omentectomy and lymphadenectomy to minimize aggressive procedures.

Adjuvant treatment in USC/UCCC is highly individualized [20]. Chemotherapy is recommended by the NCCN, and observation is only adopted in patients with stage IA disease who have no residual tumor in their hysterectomy specimens [20]. According to the 2023 NCCN guideline, cytology results should not be taken in isolation to guide adjuvant therapy. However, the guideline did not specify the management according to certain histology. Our finding was in accordance with the NCCN recommendation. In our study, we did not find a survival benefit from adjuvant chemotherapy in either positive or negative cytology patients. Chemotherapy is routinely used in our institution but may be omitted in patients with severe medical comorbidities or those who refuse additional treatment. We did not find a significant therapeutic benefit of chemotherapy on PFS or OS in patients with stage I USC/UCCC. Adding vaginal brachytherapy is preferred, but whole pelvic EBRT is no longer recommended [20]. Clinical trials are warranted in making adjuvant treatment strategies.

## 5. Conclusions

This section is not mandatory but can be added to the manuscript if the discussion is unusually long or complex.

## 6. Patents

The result of our study showed that malignant peritoneal cytology was an independent prognostic factor for inferior survival outcomes in patients with stage IA USC/UCCC. Patients with malignant peritoneal cytology had more peritoneal recurrence and a shorter time to relapse. Diagnostic hysteroscopy was associated with a worse prognosis in patients with malignant cytology.

## Figures and Tables

**Figure 1 jcm-12-01609-f001:**
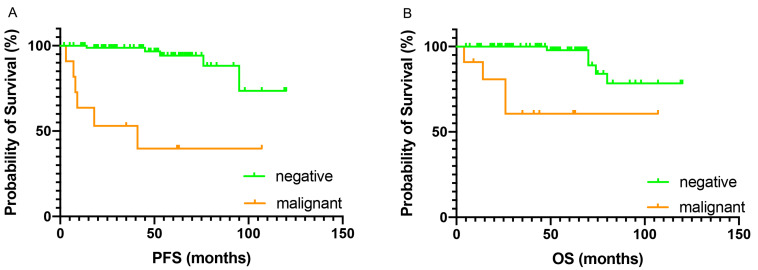
Kaplan-Meier curves comparing survival outcomes in patients who had malignant cy-tology versus negative cytology. (**A**) progression free survival (PFS); (**B**) overall survival (OS).

**Figure 2 jcm-12-01609-f002:**
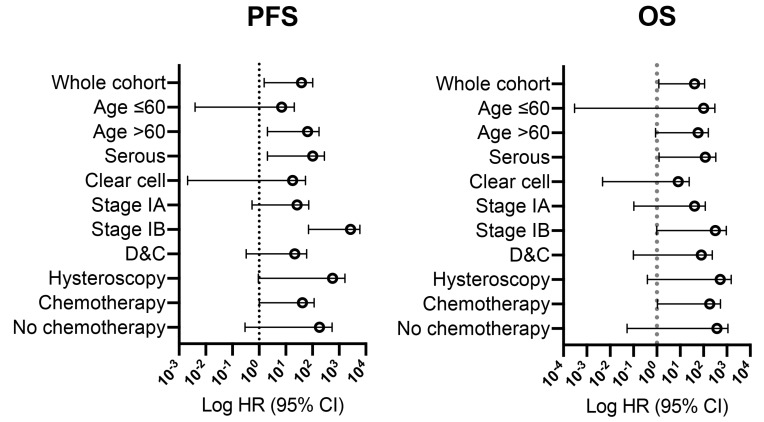
Forest plots for PFS (progression-free survival) and OS (overall survival) in sensitivity analysis of malignant cytology.

**Table 1 jcm-12-01609-t001:** Patient demographics, tumor characteristics, and treatment approaches.

	Total	Malignant Cytology	Negative Cytology	*p*-Value
	(*n* = 101)	(*n* = 11)	(*n* = 90)	
Age (years) ± SD	61.6 ± 10.0	65.5 ± 9.8	61.1 ± 9.8	0.18 ^1^
BMI ± SD	25.0 ± 4.2	24.5 ± 2.6	25.1 ± 4.4	0.67 ^1^
Maximum tumor diameter on image ± SD	2.7 ± 1.8	2.4 ± 1.5	2.8 ± 1.8	0.56 ^1^
Histology type, *n* (%)				
Serous	55 (54.5)	8 (72.7)	47 (52.2)	0.34 ^2^
Clear cell	46 (45.5)	3 (27.3)	43 (47.8)	
Stage, *n* (%)				
IA	77 (76.2)	7 (63.6)	70 (77.8)	0.29 ^2^
IB	24 (23.8)	4 (36.4)	20 (22.2)	
Grade, *n* (%)				
1	28 (27.7)	3 (27.3)	25 (27.8)	0.88 ^2^
2	11 (10.9)	1 (9.1)	10 (11.1)	
3	54 (53.5)	6 (54.5)	48 (53.3)	
NA	8 (7.9)	1 (9.1)	7 (7.8)	
Pathologic tumor size, *n* (%)				
≤1 cm	23 (22.8)	2 (18.2)	21 (23.3)	0.64 ^3^
1.1–2 cm	25 (24.8)	4 (36.4)	21 (23.3)	
>2 cm	53 (52.5)	5 (45.5)	48 (53.3	
DOI, *n* (%)				
Submucous	30 (29.7)	3 (27.2)	27 (30.0)	0.39 ^3^
<1/2 myometrium	47 (46.5)	4 (36.4)	43 (47.8)	
≥1/2 myometrium	24 (23.8)	4 (36.4)	20 (22.2)	
LVSI, *n* (%)				
No	91 (90.1)	10 (90.9)	81 (90.0)	1.00 ^2^
Yes	10 (9.9)	1 (9.1)	9 (10.0)	
Diagnostic surgery, *n* (%)				
D&C	51 (50.5)	6 (54.5)	45 (50.0)	1.00 ^2^
Hysteroscopy	50 (49.5)	5 (45.5)	45 (50.0)	
Surgery route, *n* (%)				
Open	63 (62.4)	6 (54.5)	57 (63.3)	0.74 ^2^
MIS	38 (37.6)	5 (45.5)	33 (36.7)	
Omentectomy, *n* (%)				
No	15 (14.9)	1 (9.1)	14 (15.6)	1.00 ^2^
Yes	86 (85.1)	10 (90.9)	76 (84.4)	
Appendectomy, *n* (%)				
No	65 (64.4)	8 (72.7)	57 (63.3)	0.74 ^2^
Yes	36 (35.6)	3 (27.3)	33 (36.7)	
Lymphadenectomy, *n* (%)				
No	7 (6.9)	1 (9.1)	6 (6.7)	0.56 ^2^
Yes	94 (93.1)	10 (90.9)	84 (93.3)	
Postop therapy, *n* (%)				
Chemotherapy	65 (64.4)	8 (72.7)	57 (63.3)	0.74 ^2^
Radiation therapy	13 (12.9)	2 (18.2)	11 (12.2)	0.49 ^2^

^1^ Wilcoxon rank sum *p*-value; ^2^ Fisher’s exact *p*-value; ^3^ chi-square *p*-value; DOI, depth of invasion; LVSI, lymph-vascular space invasion; D&C, dilation and curettage; MIS, minimally invasive surgery.

**Table 2 jcm-12-01609-t002:** Recurrence data.

	Malignant Cytology (*n* = 11)	Negative Cytology (*n* = 90)	*p*-Value
Total *n* (%)	6 (54.5)	5 (5.6)	<0.001 ^1^
Peritoneal (%)			0.036 ^1^
Vaginal cuff	1 (16.7)		
Pelvis	2 (33.3)	1 (20)	
Abdomen	2 (33.3)		
Lymph node (%)		2 (40)	
Hematogenous * (%)	1 (16.7)	2 (40)	
Months to relapse median (range)	13 (3–41)	38 (15–95)	0.022 ^2^

^1^ Chi-square *p*-value; ^2^ Fisher’s exact *p*-value; * all the hematogenous metastases were recurrence in lung.

**Table 3 jcm-12-01609-t003:** Univariate and multivariate analysis of potential prognostic risk factors in patients with stage I USC and UCCC.

	5-Year PFS					5-Year OS				
	Univariate			Multivariate		Univariate			Multivariate	
Variable	%	HR (95%CI)	*p*	HR	*p*	%	HR (95%CI)	*p*	HR	*p*
Cytology Malignant Negative	39.8 94.2	12.6 (1.54–103)	<0.001	20.9	<0.001	60.6 97.8	9.66 (1.21–112)	<0.001	56.2	0.001
Age >60 ≤60	81.5 97.1	3.75 (1.15–12.3)	0.064	2.00	0.459	88.8 100	6.95 (1.88–25.7)	0.030	6.64	0.173
Histology USC UCCC	83.2 93.3	4.56 (1.40–14.9)	0.031	0.20	0.096	91.0 95.8	2.52 (0.68–9.39)	0.166	0.19	0.162
Stage IA IB	89.0 84.8	0.51 (0.10–2.55)	0.303			98.4 74.8	0.08 (0.01–0.52)	0.009	10.2	0.021
Grade 1 2, 3	95.9 77.9	0.14 (0.04–0.48)	0.003	0.35	0.110	94.4 92.2	0.17 (0.04–0.65)	0.010	0.52	0.058
Size ≤2 cm >2 cm	86.2 89.5	0.534 (0.16–1.74)	0.306			97.4 89.8	0.500 (0.13–1.87)	0.303		
DOI <1/2 ≥1/2	89.1 83.5	0.492 (0.10–2.51)	0.280			96.0 81.9	0.360 (0.58–2.21)	0.128		
LVSI No Yes	87.0 88.8	0.983 (0.12–7.79)	0.98			94.0 87.5	0.673 (0.58–7.74)	0.704		
Diagnostic surgery D&C Hysteroscopy	89.8 86.1	0.574 (0.17–1.91)	0.347			97.687.6	0.309 (0.08–1.19)	0.072		
Surgery route Open MIS	89.6 85.2	0.648 (0.17–2.45)	0.467			98.1 82.0	0.354 (0.07–1.71)	0.090		
Omentectomy No Yes	61.1 92.1	2.814 (0.29–27.2)	0.159			46.1 95.9	7.96 (0.50–127)	0.015	0.02	0.003
Appendectomy No Yes	85.4 92.5	2.121 (0.75–6.97)	0.247			90.9 96.7	1.253 (0.34–4.63)	0.726		
Lymphadenectomy No Yes	80.0 88.5	3.500 (0.10–127)	0.198			NA 96.2		NA		
Chemotherapy No Yes	81.8 91.0	1.225 (0.34–4.35)	0.745			90.7 94.5	2.506 (0.62–10.0)	0.151		
Radiation therapy No Yes	87.2 92.8	1.369 (0.22–8.41)	0.762			93.6 92.8	0.993 (0.12–7.99)	0.995		

D&C, dilation and curettage.

## Data Availability

Data and materials are available from the corresponding author on reasonable request.

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
