# Peer review of "Prognostic Value of Peritoneal Cytology in Stage I Serous and Clear Cell Carcinoma of the Endometrium"

_jcm, 2023, doi:10.3390/jcm12041609_

Round 1

Reviewer 1 Report

This is a well-written paper investigating the relation of malignant peritoneal cytology and survival outcomes in patients who underwent primary staging surgery for stage I uterine serous (USC) or clear cell carcinoma (UCCC). The authors demonstrated that stage I USC or UCCC patients with malignant peritoneal cytology had higher recurrence and inferior survival. I would make the following minor comments regarding the paper:

1.      As the authors demonstrated in the Discussion, the high rate (about 77%) of patients who underwent postoperative adjuvant treatment can add heterogeneity to the results. The authors need to show the characteristics of patients who underwent postoperative adjuvant treatment.

2.      About 50% of patients underwent hysteroscopy for diagnosis of EC. As the authors described in the page 7 line 211, the intrauterine pressure during hysteroscopy could lead to tumor cell reflux and intra-abdominal dissemination. What was the reason for the high rate of hysterescopy?

Author Response

Thank you very much for reviewing our manuscript.  We greatly appreciate the reviewer’s comments and suggestions.  We have carried out the edits as suggested and revised the manuscript accordingly.  Please find attached our response to the reviewer’s questions.  We hope you find our responses satisfactory.

Point 1.      As the authors demonstrated in the ‘Discussion’, the high rate (about 77%) of patients who underwent postoperative adjuvant treatment can add heterogeneity to the results. The authors need to show the characteristics of patients who underwent postoperative adjuvant treatment.

Response 1: Thank you for your suggestion.  This is a very important point in this retrospective study.  We do concern the adjuvant radiation and chemotherapy may add the confondings into the results.  So we compared the rate and added the characteristics of patients who underwent postoperative treatment in result part in revised manuscript (line 132: “The prognostic factors in patients underwent adjuvant chemotherapy were similarly distributed between the cytology positive group and negative group: in cytology positive group, there were 6 patients diagnosed with serous cancer and 2 patients of clear cell cancer, compared to 34 and 23 patients in cytology negative group, respectively (p = 0.65); there were 5 patients had tumor size less than 2cm and 3 patients had tumor size over 2cm in cytology positive group, as compared to 35 and 22 in cytology negative group, re-spectively (p = 0.77); there were 3 patients had deep myometrium invasion in cytology positive group, as compared to 13 patients in cytology negative group, respectively (p = 0.64). Thus we consider the basic characteristics between the two groups were balanced.”)

    Since the two groups have similar rate of adjuvant chemo and radiation therapy, and the characteristics are balanced.  In addition, in the multivariate analysis, adjuvant chemotherapy and radiation therapy were not associated with survival outcomes, we consider the outcomes were comparable between the two goups. 

Point 2.      About 50% of patients underwent hysteroscopy for diagnosis of EC. As the authors described in the page 7 line 211, the intrauterine pressure during hysteroscopy could lead to tumor cell reflux and intra-abdominal dissemination. What was the reason for the high rate of hysterescopy?

Response 2: Thank you for your comments. Since dialation and curretage had high rate of false negative rate and endometrial biopsy had low sensitivity in diagnose endometrial cancer (ref. van Hanegem N, et al. The accuracy of endometrial sampling in women with postmenopausal bleeding: a systematic review and meta-analysis. Eur J Obstet Gynecol Reprod Biol. 2016 Feb;197:147-55.) In our institution, we have adopt hysterectomy as golden standard for over 10 years. It is consider oncologically safe since the cytology was no longer a factor for staging. It is safe in type I EC patients and now we are more cause in patients highly suspected of type II EC. We revised the manuscript as: line 229, “For patients with postmenopausal bleeding, hysteroscopy was considered more sensitive and accurate as compared to endometrial sampling and blind dilation & curettage.[16] It was reported that blind dilation & curettage can miss nearly 50% of uterine cavity’s endometrium, in terms of high false negative rate. Since the prognostic value of positive cytology was minimal especially in early-stage EC and lack of evidence that hysteroscopy can cause tumor cell influx into abdominal cavity, hysteroscopy was used as golden standard in diagnosis of abnormal uterine bleeding in our institution.”

We would like to thank the reviewers for their thoughtful comments and efforts towards improving our manuscript. We hope you find our revised manuscript satisfactory for publication.

Reviewer 2 Report

Dear authors

I have carefully read your manuscript evaluating role of peritoneal cytology in Early stage Serous & Clear cell Carcinoma endometrium.

I have certain observations:

1. The background of this study is built on the premise that malignant cytology has no clear role in management of high risk histologies like Serous & Clear cell endometrial malignancy. However, this is incorrect as 2023 NCCN guidelines clearly spell out role of cytology, with adjuvant therapy indicated in positive cytology.

2. Only 10% of the patients in this retrospective study over past decade had positive cytology, and this group is the focus of this study. However, no separate results have been described for this study group. What was the extent of adjuvant treatment administered to these patients? What were other pathologic factors in this subset of patients? It is highly likely that other factors (pathology & treatment) influenced the outcome of this sub-group and has contributed more bias.

Author Response

Thank you very much for reviewing our manuscript.  We greatly appreciate the reviewer’s comments and suggestions.  We have carried out the edits as suggested and revised the manuscript accordingly.  Please find attached our response to the reviewer’s questions.  We hope you find our responses satisfactory.

Point 1. The background of this study is built on the premise that malignant cytology has no clear role in management of high risk histologies like Serous & Clear cell endometrial malignancy. However, this is incorrect as 2023 NCCN guidelines clearly spell out role of cytology, with adjuvant therapy indicated in positive cytology.

Response 1: Thank you for your recommendation. This retrospective study was started in 2020 and completed in 2022 so the initiative was clear at that time. We reviewed the 2023 NCCN guidelines and found it admitted positive cytology as an adverse risk facto. It does not recommand to take cytology results in isolation to guide adjuvant therapy. Our finding was in accordance to the NCCN recommendation, since we did not find survival benefit from chemotherapy in early stage USC and UCCC regardless of cytology status. We added it to the revised manuscript(line 286: “The 2023 NCCN guideline, cytology results should not be taken in isolation to guide adjuvant therapy. However, the guideline did not specify the management according to certain histology. Our finding was in accordance to the NCCN recommendation. In our study we did not find survival benifit from adjuvant chemotherapy in either positive or negative cytology patients.”)    

Point 2. Only 10% of the patients in this retrospective study over past decade had positive cytology, and this group is the focus of this study. However, no separate results have been described for this study group. What was the extent of adjuvant treatment administered to these patients? What were other pathologic factors in this subset of patients? It is highly likely that other factors (pathology & treatment) influenced the outcome of this sub-group and has contributed more bias.

Response 2: Thank you for your comments. We added the information of cytology positive patients in revised manuscript (line 119 “In the 11 patients with malignant peritoneal cytology, 8 patients were diagnosed with serous carcinoma and 3 patients were diagnosed with clear cell carcinoma. Among the 8 patients with serous carcinoma, 4 patients underwent open surgery and 4 patients un-derwent MIS; 5 of them received adjuvant chemotherapy of 3 cycles of TC. Among the 3 patients with clear cell carcinoma, 2 patients received open surgery and 1 received MIS; they all received adjuvant chemotherapy of 3 cycles of TC.”) and compared the basic characteristics of the two groups in table 1.

    We compared the rate and added the characteristics of patients who underwent postoperative treatment in result part in revised manuscript (line 132: “The prognostic factors in patients underwent adjuvant chemotherapy were similarly distributed between the cytology positive group and negative group: in cytology positive group, there were 6 patients diagnosed with serous cancer and 2 patients of clear cell cancer, compared to 34 and 23 patients in cytology negative group, respectively (p = 0.65); there were 5 patients had tumor size less than 2cm and 3 patients had tumor size over 2cm in cytology positive group, as compared to 35 and 22 in cytology negative group, re-spectively (p = 0.77); there were 3 patients had deep myometrium invasion in cytology positive group, as compared to 13 patients in cytology negative group, respectively (p = 0.64). Thus we consider the basic characteristics between the two groups were balanced.”) The two groups have similar rate of adjuvant chemo and radiation therapy, and the characteristics are balanced.  In addition, in the multivariate analysis, adjuvant chemotherapy and radiation therapy were not associated with survival outcomes, we consider the outcomes were comparable between the two goups. 

We would like to thank the reviewers for their thoughtful comments and efforts towards improving our manuscript. We hope you find our revised manuscript satisfactory for publication.

Round 2

Reviewer 2 Report

Dear authors

I am satisfied with the revisions